# No evidence that omission and confirmation biases affect the perception and recall of vaccine-related information

Ángel V. Jiménez[ID][1,2,3]*, Alex Mesoudi[3], Jamshid J. Tehrani[1,2]

1 Durham Cultural Evolution Research Centre, Department of Anthropology, Durham University, Durham, United Kingdom, 2 Conspiracy Theories in Health Special Interest Group, Wolfson Research Institute for Health and Wellbeing, Durham University, Durham, United Kingdom, 3 Human Behaviour and Cultural Evolution Group, Department of Biosciences, University of Exeter, Penryn, United Kingdom

* aj419@exeter.ac.uk

**Data Availability Statement:** All relevant data are within the paper and its Supporting Information files.

## Abstract

Despite the spectacular success of vaccines in preventing infectious diseases, fears about their safety and other anti-vaccination claims are widespread. To better understand how such fears and claims persist and spread, we must understand how they are perceived and recalled. One influence on the perception and recall of vaccination-related information might be universal cognitive biases acting against vaccination. An *omission bias* describes the tendency to perceive as worse, and recall better, bad outcomes resulting from commissions (e.g. vaccine side effects) compared to the same bad outcomes resulting from omissions (e.g. symptoms of vaccine preventable diseases). Another factor influencing the perception and recall of vaccination-related information might be people's attitudes towards vaccines. A *confirmation bias* would mean that pre-existing pro-vaccination attitudes positively predict perceptions of severity and recall of symptoms of vaccine preventable diseases and negatively predict perceptions of severity and recall of vaccine side effects. To test for these hypothesized biases, 202 female participants aged 18–60 ($M = 38.15$, $SD = 10.37$) completed an online experiment with a between-subjects experimental design. Participants imagined that they had a 1-year old child who suffered from either vaccine side effects (Commission Condition) or symptoms of a vaccine-preventable disease (Omission Condition). They then rated a list of symptoms/side effects for their perceived severity on a 7-point Likert scale. Finally, they completed a surprise recall test in which they recalled the symptoms/side effects previously rated. An additional scale was used to measure their attitudes towards vaccines. Contrary to the hypotheses, perceptions of severity and the recall of symptoms/side effects were not associated with experimental condition, failing to support the omission bias, nor did they interact with attitudes towards vaccines, failing to support the confirmation bias. This cast doubt on the possibility that the spread of anti-vaccination claims can be explained by these particular universal cognitive biases.

**Funding:** This work was supported by the Leverhulme Trust Research Project Grant RPG-2016-122 (https://www.leverhulme.ac.uk/) to AM. The funder had no role in study design, data collection and analysis, decision to publish, or preparation of the manuscript.

**Competing interests:** The authors have declared that no competing interests exist.

## Introduction

Ever since vaccination was developed in the late 18th century, it has been accompanied by scepticism and opposition. Early resistance to the first vaccine, against smallpox, was motivated by fears towards a practice that conflicted with folk intuitions and religious beliefs [1–3]. Smallpox vaccination entailed the introduction of an alien substance into the human bloodstream, which was seen as a violation of bodily purity and as the cause of, rather than the cure for, diseases [3]. Moreover, the mechanism by which vaccination against smallpox worked was counterintuitive as it implied that the intentional infection with one disease (cowpox) could prevent the future development of a related but different disease (smallpox) [2]. Furthermore, smallpox vaccination involved the mixture of a substance coming from a cow with human blood, which was considered anti-natural and anti-Christian in Europe [2] and was also disturbing for Hindus in colonial India [1].

More recently, specific vaccines (e.g. the MMR vaccine) have been the targets of opposition and have often been falsely linked to idiopathic medical conditions such as autism, neurological disorders and allergies [4–6]. Anti-vaccination messages recurrently emerge and spread through the mass media [7, 8], social media [9] and on anti-vaccination websites [10–14]. This spread of anti-vaccination messages affects people's decisions regarding whether to vaccinate or not [15, 16]. As a consequence of the erosion of herd immunity and the formation of clusters of unvaccinated individuals [17], outbreaks of vaccine-preventable infectious diseases still occur in developed countries, sometimes leading to fatal consequences.

The recurrence and potency of social fears about the safety of vaccines and the widespread diffusion of anti-vaccination information, despite the spectacular contribution of vaccines in preventing diseases, calls for a scientific explanation. Miton and Mercier [18] recently argued that two characteristics of human cognition—disgust towards potential contaminants and an omission bias—make vaccination counterintuitive. This counter-intuitiveness makes anti-vaccination messages "cognitively attractive" and likely to spread (see [19] for a general discussion on how scientific and pseudoscientific ideas spread). In the present article we empirically test the role of one of these cognitive factors, the omission bias, i.e. the tendency to consider bad outcomes resulting from a commission (e.g. side effects of a vaccine) as worse that the same bad outcomes resulting from an omission (e.g. symptoms of a vaccine-preventable disease). While there is some evidence for the existence of omission bias in the context of vaccination (see next section), our study is novel in extending the scope of application of the omission bias to perceptions of the severity of vaccine-related symptoms and side effects and their recall. Consequently, we predict that people should rate as more severe, and retain better in memory, vaccine side effects over symptoms of a vaccine-preventable disease, where the symptoms and the side effects are otherwise identical. These perceptual and memory effects relating to vaccine-related information (symptoms and side effects) would play a key role in the retention and transmission of anti-vaccination messages and opinions. Our study is one of the first experiments exploring the effects of cognitive biases in the *recall* of vaccine-related information (see also [20, 21]).

### 1.1.- The omission bias in vaccine decisions

A substantial body of literature maintains that humans have a general tendency to consider the bad outcomes resulting from an action (commission) as worse than the same bad outcomes resulting from a lack of action (omission), even when the bad outcomes resulting from an omission affect a greater number of people or have a higher probability of occurrence (e.g. [22]). The basic paradigm for studying the omission bias applied to vaccine decisions was developed by Ritov and Baron [23]. Participants are prompted to imagine a scenario in which

an infectious disease kills 10 out of 10,000 non-vaccinated children but a new vaccine confers immunity against this disease. However, the side effects of this vaccine can result in death with a certain probability. Participants have to decide whether or not to vaccinate their child when the probability of death by the vaccine side effects is 0, 1, 2, 3, 4, 5, 6, 7, 8, 9 or 10 out of 10,000 vaccinated children. The omission bias is detected when participants prefer not to vaccinate when the probability of dying from vaccine side effects is lower than the probability of dying from the vaccine-preventable infectious disease. Multiple experiments have found that many people prefer not to vaccinate themselves or their children when the risks of suffering from a vaccine-preventable disease are higher than the risks of suffering from a vaccine reaction, where the severity of the disease and the vaccine reaction are equal [23–31]. Importantly, the omission bias in hypothetical vaccine decisions positively predicts actual vaccine behaviour in both retrospective [24, 26, 28] and prospective studies [31].

One of the cognitive mechanisms that might explain the omission bias is anticipated regret. Some authors (e.g.[24]) have suggested that bad outcomes resulting from actions elicit considerably more regret than the same bad outcomes resulting from a lack of action. The higher level of regret after bad outcomes resulting from commissions is probably motivated by a greater perception of the causality of commissions than of omissions [22]. This often makes people judge commissions with negative consequences as morally wrong [22]. Nevertheless, anticipated regret can also be experienced towards bad outcomes resulting from omissions [32–35] and this might partly explain high vaccination rates in post-industrial societies.

From an evolutionary viewpoint, reliable estimations of the probabilities of risks were not available during the greatest part of human history. Hence, it is plausible that natural selection equipped humans with some cognitive biases (e.g. the omission bias) that, while technically or statistically incorrect, maximised their bearers' chances of survival and reproduction in ancestral environments. According to error management theory [36], cognitive biases evolved as a result of the asymmetry between the costs of false positive and false negative errors in some areas of human life in our evolutionary past. In this sense, the greater risks associated with some actions (e.g. consuming an unknown toxic plant) in comparison with the lower risks associated with inactions (e.g. avoiding consuming an unknown non-toxic plant) might have prepared human psychology to consider bad outcomes resulting from commissions as worse than those resulting from omissions. While this bias may have been an adaptive cognitive short-cut in our ancestors, the persistence of this bias may be maladaptive in modern environments when people have access to reliable information that the asymmetry between false positives and false negatives does not occur.

## 1.2.- The omission bias in severity ratings

In the present study we build on previous work that has tested for omission bias in the context of vaccination. While the research reviewed above has shown that omission bias may affect people's decisions about the acceptability of different risk probabilities, concerns have been raised over the adequacy of the kind of probability judgements described above as used by Ritov and Baron [23]. As Connolly and Reb [27] argue, such a method "requires the subject to make complex tradeoffs of utility and probability" (p.189), amongst other problems. A more realistic method, introduced by Connolly and Reb, is to ask participants to rate the severity of symptoms caused by commissions and omissions. Here, the omission bias would occur when people assess the same symptoms as more severe when they are the result of vaccine side effects (commission) than when they are the result of symptoms of a vaccine-preventable disease (omission). Contrary to the operation of an omission bias, Connolly and Reb found that the majority (60%) of participants in their study rated disease symptoms as equally serious as

vaccine side-effects, with only 8% rating side-effects as more serious in line with omission bias. However, the stated symptoms in that study were rather vague (e.g. "feeling horrible") and rated together rather than separately. A subsequent study by Brown, Kroll [37] allowed participants to assign specific symptoms to 'mild', 'moderate' and 'severe' diseases and vaccine reactions. They found that for 9 out of 13 symptoms/reactions, more participants assigned the symptom/reaction to a milder category when associated with a disease than to a vaccine side-effect, which the authors interpreted as evidence for omission bias. Given this contradictory evidence in the literature, our first aim here was to provide a further test of whether omission bias affects severity ratings for vaccine-related symptoms and side-effects.

## 1.3- The omission bias in recall

Our second aim is to test whether omission bias also differentially affects memory for symptoms and side effects. If natural selection has provided human psychology with a tendency to consider as worse the bad outcomes resulting from commissions than omissions, it is plausible that natural selection would have also made people pay more attention to and recall better bad outcomes resulting from commissions than bad outcomes resulting from omissions. If this is true, vaccine side effects (commissions) should be better recalled than symptoms of a vaccine-preventable infectious disease (omissions). This prediction relates to research on the adaptive basis of memory and cultural transmission. Several studies have shown that adaptively-relevant information, or information placed within an adaptive context, is better recalled and transmitted along chains of participants than adaptively-neutral information, such as information related to survival [38], social interactions [39, 40] and disgusting stimuli [41]. No previous research to our knowledge has tested the effects of omission bias on recall of vaccine-related information, but this potentially plays a major role in the spread of vaccine (mis-)-information in society.

## 1.4.- The confirmation bias

A final aim is to compare the omission bias against another potential bias relating to congruity between presented information and pre-existing attitudes, known as confirmation bias. The omission bias predicts that vaccine side effects are rated as more severe and are recalled better than symptoms of a vaccine-preventable disease in a manner that is relatively independent of people's attitudes towards vaccination. However, an alternative hypothesis is that people's assessments of severity and recall depend on their general attitudes towards vaccination. That is, people with pro-vaccination attitudes would rate as more severe and recall better symptoms of a vaccine-preventable disease than vaccine side effects, while people with anti-vaccination attitudes would rate as more severe and recall better vaccine side effects than symptoms of a vaccine-preventable disease. The confirmation bias follows from research showing that people are more likely to attend to and recall information that supports or confirms their pre-existing attitudes on issues such as gun control [42], as well as the larger literature on motivated reasoning and cognition [43]. As others have done for omission bias, Mercier and Sperber [43, 44] suggest an evolutionary basis for confirmation bias: if the adaptive function of reasoning is to act as a process of social argumentation, negotiation and persuasion, then selectively perceiving and recalling information that supports one's position makes human communication more effective and mutually advantageous. However, Altay and Mercier [21] did not find that participants with pro-vaccination attitudes recall better perceived pro-vaccination messages than perceived anti-vaccination information, failing to support the confirmation hypothesis as applied to vaccination. Together with that study, our study is one of the first to test whether

the confirmation bias plays a role in the transmission of vaccine-related information, and the first to directly compare omission and confirmation biases.

## 1.5.- Hypotheses

We specify two pairs of hypotheses, one pair for the dependent measure 'severity ratings' and another pair for the dependent measure 'recall'.

Hypotheses about severity ratings:

- H1a (omission bias hypothesis): Vaccine side effects (commissions) will be assessed as more severe than symptoms of a vaccine-preventable disease (omissions), irrespective of people's vaccination attitudes.

- H1b (confirmation bias hypothesis): People with anti-vaccination attitudes will rate as more severe the vaccine side effects (commissions), while people with pro-vaccination attitudes would rate as more severe the symptoms of a vaccine-preventable disease (omissions).

Hypotheses about recall:

- H2a (omission bias hypothesis): The bad outcomes resulting from vaccine side effects (commissions) will be better recalled than bad outcomes resulting from a vaccine-preventable disease (omissions).

- H2b (confirmation bias hypothesis) People with anti-vaccination attitudes will recall to a greater extend vaccine side-effects (commissions), while people with pro-vaccination attitudes will recall to a greater extent symptoms of a vaccine-preventable disease (omissions).

## Methods

### 2.1.- Ethical statement

This study was approved by the Ethics Committee at the Department of Anthropology of University of Durham on 24[th] June 2016. Participants were informed of the procedure before the experiment began. They were also informed that their participation was anonymous and confidential and that they could withdraw from participating in the study at any time by simply closing the browser without having to give any kind of explanation. After reading this, participants provided their consent to participate in the study and to use their anonymous responses for scientific publications by ticking a box.

### 2.2.- Participants

Participants were recruited through Prolific (www.prolific.ac) following the procedure stated in the preregistration, which can be found on the Open Science Framework (OSF) website (https://osf.io/gebc7/). This procedure entailed using pre-screening filters to select participants who had not previously participated in any of our vaccine-related experiments, had an approval rate of 90% or above, were aged 18–60 years, spoke English as a first language and had British or American nationality. These characteristics were assumed to increase the chances of collecting responses provided by participants who understood the content of the information and paid attention to the materials. In addition, participants were pre-screened by their gender and attitudes towards vaccination. Chandler and Paolacci [45] have shown that participants can lie by self-reporting rare conditions (e.g. having a child with autism) in order to get access to more studies. As did they, we found that men showed a greater tendency to report rare conditions

(anti-vaccination attitudes in our case) than women [45]. This threatens the validity of the present study. Consequently, we decided to include only women in the experiment, as their pre-screened anti-vaccination attitudes were more reliable than the ones reported by men. This comes with the obvious cost that we can only make inferences about women's ratings and recall, and not men's.

We selected 50% participants who self-reported pro-vaccination attitudes and 50% who self-reported anti-vaccination attitudes. For pre-screening vaccination attitudes, participants indicated their agreement with the following item on a scale from 1 (totally disagree) to 7 (totally agree): "*If I had a baby to care for, I would want him/her to get all the recommended immunizations*" [46]. Participants who agreed (ratings from 5 to 7) were considered to have pro-vaccination attitudes and participants who disagreed (ratings from 1 to 3) were considered to have anti-vaccination attitudes.

Consequently, we collected responses from 261 participants (260 female, 1 male) aged 18–60 ($M = 38.15$, $SD = 10.37$) with American ($N = 43$), British ($N = 216$) or other nationality ($N = 2$), being all but one English native speakers. After excluding participants who did not meet the inclusion criteria of being (i) female, (ii) an English native speaker, (iii) aged 18–60 and (iv) reading the experimental manipulation at a pace greater than 400 words per minute [47], the final sample comprised 202 female participants (38 American, 164 British) aged 18–60 ($M = 38.61$, $SD = 10.39$). The data was collected on 5[th] March and 31[st] March 2018.

### 2.3.- Materials

The experiment was programmed and administered using Gorilla (www.gorilla.sc/about). A list of 24 symptoms were used for obtaining measures of severity ratings, reaction times and recall. All these symptoms were selected from a larger list of 54 symptoms. 58 participants (22 males, 36 females) aged 19–76 ($M = 38.10$, $SD = 14.10$) recruited through Crowdflower rated each symptom for familiarity and severity on a 7-point Likert scale from 1 (very unfamiliar/mild) to 7 (very familiar/severe). All the symptoms selected for the experiment were highly familiar ($M > 4.5$) and their severity had a large variability ($SD > 1.3$) for this independent sample of participants (Table 1). See Supplementary Materials A and B in S1 File for further details.

Two different texts were created as processing conditions. The texts were adapted from Connolly and Reb [27] who themselves adapted the scenarios from Ritov and Baron [23] and Asch, Baron [24]. The greatest part of the information was similar across the two texts. The texts varied in only one aspect: one text prompted the participants to imagine that they had a one-year-old child who suffered from a set of symptoms as a consequence of a vaccine-preventable infectious disease (Omission Condition), while the other text prompted the participants to imagine that the symptoms were the consequence of the vaccine side effects (Commission Condition):

**Table 1. List of symptoms/side effects using in the experiment.**

| LIST OF SYMPTOMS/SIDE EFFECTS | |
|---|---|
| Symptoms selected for the experiment | crying, shivering, chills, swelling, cough, nausea, vomiting, diarrhoea, headache, earache, wheezing, abdominal pain, muscle ache, loss of appetite, soreness, hoarseness, itchy eyes, toothache, light-headedness, dry mouth, weight gain, anxiety, insomnia, bleeding. |
| Symptoms selected for the practice | irritability, fever, rash, sneezing, dizziness |

*Imagine that, in the country in which you live, there have been several outbreaks of a new infectious disease. The disease can cause severe illness to children under three. Only a small number of children exposed actually catch this disease, but for those who do it is often quite severe.*

*A vaccine for this disease has been developed and tested. The vaccine eliminates any possibility of the vaccinated child getting the disease. The vaccine, however, can sometimes cause side effects that are very similar to the symptoms of the disease. Fortunately, these unpleasant side effects are rare. In fact, the risk of a vaccinated child getting the unpleasant side effects is about as low as the chance of a non-vaccinated child getting the severe symptoms of the disease.*

*Imagine that you have one child, a one-year old. Suppose you decided* **to vaccinate** [**commission condition**] */ not to vaccinate* [**omission condition**]. *Unfortunately, your child is one of those who suffer from the* **vaccine side effects** [**commission condition**] */ symptoms of the disease* [**omission condition**].

An additional sample of 53 participants (28 males, 24 females and 1 other; 39 British and 14 Americans) aged 20–58 years (*M* = 36.26, *SD* = 11.11), who were recruited through Prolific, rated these texts for their quality and their elicitation of different emotions (disgust, regret, fear, anger, sadness, joy, compassion, surprise and confusion). These ratings did not differ across the texts. See Supplementary Materials C and D in S1 File for further details.

Two items were included to measure attitudes towards vaccination ("*If I had a baby to care for, I would want him/her to get all the recommended immunizations*" and "*I believe that scheduled immunizations are safe for children*"). These items were previously used by Browne, Thomson [46], who adapted the items themselves from Gust, Strine [48]. A 7-point Likert scale was used to collect the participants' responses (1 = totally disagree, 7 = totally agree) on these three items. The scores of the two items about vaccination attitudes were summed together, divided by two and converted into a scale from -3 to +3. The internal consistency of the scale was high (Cronbach's $\alpha$ = 0.95).

## 2.4.- Design and procedure

A between-subjects design with two conditions (*omission* vs *commission*) was employed. After providing consent participants were randomly assigned to one of the experimental conditions and then the experiment began. The procedure was similar to the *survival processing paradigm* [38]. First, participants were shown one of the two processing condition scenarios (i.e. prepared texts). Then the participants were told that they would be shown a list of symptoms/side effects that their child might have as a consequence of their decision to vaccinate/not to vaccinate. The task consisted of rating the severity of the 24 symptoms/side effects on a scale from 1 (very mild) to 7 (very severe). To give their answers, participants had to move the mouse on the screen and click on a number (see https://gorilla.sc/openmaterials/58919).

Before the experiment started, they practiced the experimental procedure with a list of five symptoms/side effects to ensure that the participants fully understood the task (Table 1). After completing the practice, the participants had the opportunity to read the processing conditions and instructions again. Then, they rated the severity of 24 symptoms/side effects. The symptoms/side effects appeared randomly at the center of the screen. The participants had up to five seconds to rate the severity of each symptom/side effect. If participants responded before 5 seconds, the symptoms/side effect disappeared and the next trial started. Between trials a fixation point "+" appeared at the center of the screen for 500 milliseconds.

The next stage of the experiment entailed the completion of the distractor task that involved solving easy sums (e.g. 2+3) for 2 minutes. Afterwards, the instructions for the surprise free recall test appeared on the screen. The participants were prompted to recall the previously presented symptoms/side effects. Participants had 3 minutes to complete this task. A countdown

timer appeared on the screen. When the countdown reached zero, the participants were auto-matically redirected to the last phase of the experiment, in which they had to complete the scale about attitudes towards vaccination and report their demographic data (gender, age, nationality, native language, education and number of children).

## Results

Statistical analyses were preregistered on the OSF website (https://osf.io/gebc7/). In the prereg-istered protocol, we planned to test the confirmation bias hypothesis by conducting regression models with interaction terms without main effect terms (e.g. Average Severity ~ Condition: Vaccination Attitudes). However, the use of interaction terms without main effect terms is controversial and, consequently, we conducted additional analyses including both interaction and main effect terms (e.g. Average Severity ~ Condition $*$ Vaccination Attitudes). Following statistical advice, we report here these latter analyses. Nevertheless, this alternative procedure did not qualitatively change the results (see Supplementary Material E in S1 File).

### 3.1.- Attitudes towards vaccination

Although participants were pre-screened to ensure a balanced distribution of pro-vaccination and anti-vaccination attitudes, the sample was considerably skewed towards pro-vaccination attitudes ($M = 1.13$, $SD = 1.93$). As expected, participants who reported pro-vaccination atti-tudes in the pre-screening showed very positive attitudes towards vaccines in the experiment ($M = 2.52$, $SD = 0.74$). In contrast, participants with anti-vaccination attitudes in the pre-screening reported very variable attitudes towards vaccines ranging from -3 to +3 ($M = -0.29$, $SD = 1.85$). Nevertheless, the distribution of participants' attitudes towards vaccines was very similar in both conditions (Commission: $M = 1.14$, $SD = 1.99$; Omission: $M = 1.12$, $SD = 1.88$). See Fig 1.

### 3.2.- Average severity ratings

For our analyses of average severity ratings, we adopted a model comparison approach [49]. We first ran a null linear model that used a fixed intercept (i.e. the mean of the outcome vari-able) to predict each participants' average severity ratings. We compared this null model with two linear regression models directly derived from our hypotheses using Akaike's Information Criterion (AIC, see Table 2). A lower AIC by a difference of two or larger was considered evi-dence of better fit to the data.

 The first model (AIC = 492.8) tested the omission bias hypothesis in severity ratings (H1a), which predicted greater severity ratings in the commission than in the omission condition. This model included condition as the sole predictor of average severity ratings. The second model (AIC = 496.9) tested the confirmation bias hypothesis in severity ratings (H1b), which predicted a positive relationship between vaccination attitudes and average severity ratings in the omission condition and a negative relationship between these variables in the commission condition. This model included attitudes towards vaccination, average severity ratings and their interaction as predictors of average severity ratings. Contrary to H1a and H1b, none of these models had a better fit than the intercept-only null model (AIC = 492.8) (see Fig 2). Similarly, exploratory analyses of the effects of the demographic variables (i.e. nationality, age, education, having children or not) and their interaction with the experimental condition and participants' attitudes towards vaccines did not improve the model fit compared to the null model.

## Attitudes Towards Vaccines

**Fig 1. Frequencies of the different scores in the scale of vaccination attitudes split by condition from -3 (extremely anti-vaccination) to + 3 (extremely pro-vaccination).**

### 3.3- Recall

For our analyses of recall, we use the same model comparison approach as for the analyses of average severity ratings. As our outcome variable was the proportion of correctly recall symptoms/side effects, we model the relationship between the predictors and the outcome variables using logistic regression models for a proportional outcome [50]. The fit of a null model was compared with our confirmatory models (see Table 3). As for the ratings of severity, a lower AIC by a difference of two or larger was considered evidence of better fit to the data. Correct recall was assessed by comparing each of the symptoms/side effects listed by the participants with the list of symptoms/side effects for the experiment. This comparison was semantic. For instance, symptoms/side effects such as irritated eyes, sore throat and sickness were coded as equivalent to itchy eyes, hoarseness and nausea respectively. Other symptoms/side effects

**Table 2. Model Comparisons for confirmatory models to predict average severity ratings.** 'Condition' specified whether the participant received the commission or the omission scenario.

| Models for Confirmatory Analyses | AIC | dAIC | df | weight | Residual Deviance |
|---|---|---|---|---|---|
| Null | **492.8** | 0 | 2 | 0.667 | 133 |
| Condition | **492.8** | 2 | 3 | 0.246 | 133 |
| Condition X Vaccination Attitudes | **496.9** | 4.1 | 5 | 0.087 | 131.7 |

'Condition' specified whether the participant received the commission or the omission scenario.

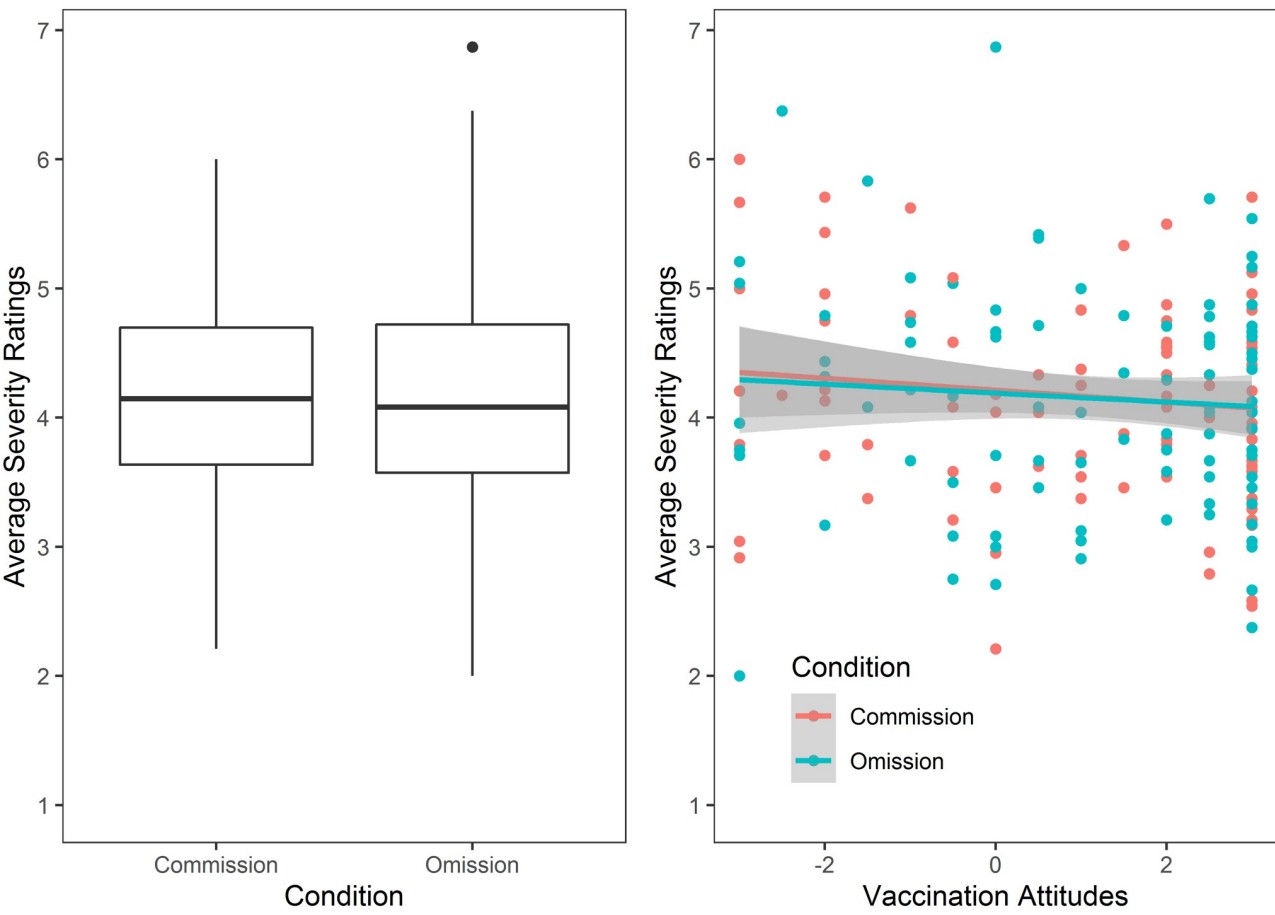

**Fig 2. Average severity ratings across vaccination attitudes in both conditions (Omission and Commission).** Left: Contrary to the omission bias hypothesis (H1a), the average severity ratings were almost identical in the Omission and Commission Conditions. Right: Contrary to the confirmation bias hypothesis (H1b), the severity ratings did not differ as a function of the interaction between condition and vaccination attitudes. The shaded area represents 95% confidence interval.

which were not semantically similar to the symptoms/side effects presented to the participants (e.g. fatigue, cold, nervousness) were ignored as errors.

First, three control models including average reaction time and average severity ratings together and separately were produced to control the possible effects of these variables on

**Table 3. Model comparisons for confirmatory models to predict correct recall.** 'Condition' specified whether the participant received the commission or the omission scenario.

| Models for Confirmatory Analyses | AIC | dAIC | df | weight | Residual Deviance |
|---|---|---|---|---|---|
| Null | 1028.2 | 3.6 | 1 | 0.06 | 321.6 |
| Reaction Time (Control Model) | 1025.2 | 0.6 | 2 | 0.269 | 316.6 |
| Condition | 1029.4 | 4.8 | 2 | 0.033 | 320.8 |
| Condition + Reaction Time | 1025.9 | 1.2 | 3 | 0.192 | 315.3 |
| Condition X Vaccination Attitudes | 1027.4 | 2.8 | 4 | 0.090 | 314.8 |
| Condition X Vaccination Attitudes + Reaction Time | 1024.7 | 0 | 5 | 0.356 | 310.1 |

'Condition' specified whether the participant received the commission or the omission scenario.

recall, and their model fit was compared to each other and to the null model. The model with only average reaction time as a predictor (AIC = 1025.2) was the only model with a better fit than the null model (AIC = 1028.2). As a consequence, this model was used as a base for the confirmatory analyses.

To test the omission bias hypothesis in recall (H2a), which predicts a greater recall of vaccine side effects (commission condition) than symptoms of a vaccine-preventable disease (omission condition), two models were produced. The first model included only condition as predictor of recall (AIC = 1029.4). The second model included additively both average reaction time and condition as predictors (AIC = 1025.9). None of these models improved the model fit of the control model that contained only average reaction time (AIC = 1025.2). To test the confirmation bias hypothesis in recall (H2b), which predicts a positive relationship between the proportion of symptoms/side effects correctly recalled and vaccination attitudes in the omission condition and a negative relationship between these variables in the commission condition, two models were produced. The first model included condition, vaccination attitudes and their interaction as predictors of recall (AIC = 1027.4). The second model also included average reaction time as a predictor (AIC = 1024.7). None of these models improved the model fit of the control model (AIC = 1025.2). Therefore, we did not find support for H2a or H2b. See Fig 3.

Exploratory analyses of the effects of the demographic variables (i.e. nationality, age, education, having children or not) and their interaction with the experimental condition and participants' attitudes towards vaccines were also conducted. Models with education as a predictor improved the fit over the control model, the best model being the one that included the additive effect of average reaction time and education (AIC = 1011.8). In this model, both education ($\beta$ = 0.23, $SE$ = 0.06) and average reaction times ($\beta$ = 0.09, $SE$ = 0.06) were positively related to the proportion of symptoms/side effects correctly recalled. Neither of these effects were predicted, nor do they relate to our hypothesised effects.

## Discussion

We found no evidence to support the omission bias hypothesis in either severity ratings (H1a) or recall (H2a) of vaccine-related symptoms and side-effects. The same physical conditions were rated as equally severe and recalled equally well when described as disease symptoms and as vaccine side-effects, contrary to our prediction that side-effects should be rated stronger and recalled better. We also found no evidence for the confirmation bias in either severity ratings (H1b) or recall (H2b). Participants' stance on vaccination had no effect on their severity ratings or recall of disease symptoms and vaccine side-effects, contrary to our prediction that pro-vaccination participants would rate more severely and recall better disease symptoms than vaccine side-effects, and anti-vaccination participants the reverse.

Our null finding with respect to omission bias contrasts with the many supportive studies reviewed in the Introduction. However, as noted above, many of these studies may suffer from methodological problems relating to the presentation of unintuitive probability information [27]. Following Connolly and Reb [27], we used a more intuitive method focusing on concrete physical symptoms and side effects rather than abstract and extreme probabilities of mortality. Like Connolly and Reb, we found no evidence for omission bias despite better controlled and more detailed presentation of symptoms and side effects. This counts against Brown et al. [37] who purported to find omission bias in severity ratings of specific symptoms and side effects. However, Brown et al.'s method is somewhat convoluted, with participants asked to create mild, moderate and severe diseases and side effects. Their statistical analysis did not correct for

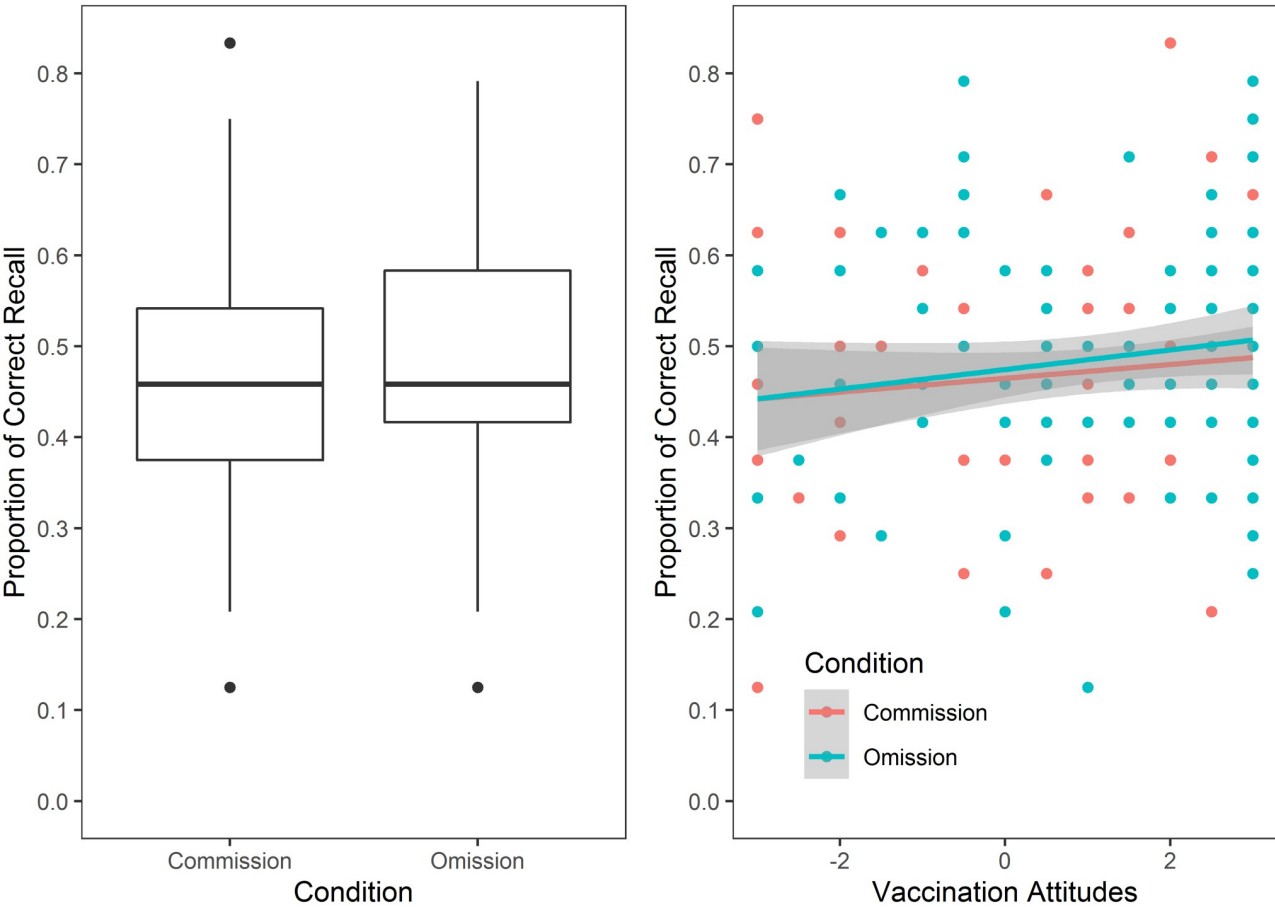

**Fig 3. Proportion of symptoms/side effects correctly recalled across vaccination attitudes in both conditions (Omission and Commission).** Left: Contrary to the omission bias hypothesis (H2a), the proportion of correct recall was similar in the Omission and Commission Conditions. Right: Contrary to the confirmation bias hypothesis (H1b), the proportion of correct recall did not differ as a function of the interaction between condition and vaccination attitudes. The shade area represent 95% confidence interval.

running multiple t-tests; when we apply Holm-Bonferroni correction to their 13 comparisons the number that are statistically significant reduced from 9 to just 6.

Like Connolly and Reb [27], then, we are sceptical of the existence of omission bias as applied to vaccination, especially in light of the results of a recent study [20]. This study use a different experimental procedure, the method of serial reproduction [51, 52], and similarly found no support for the omission bias. Therefore, we recommend re-examination of prior positive findings in the literature. In relation to evolutionary arguments for the adaptiveness of omission bias, it is possible that the selection pressure in our evolutionary past required for the evolution of the omission bias was not as strong as the selection pressure for the evolution of other types of content transmission biases such as the tendency to recall and transmit better social [39, 40, 53], emotional [41, 54, 55] and survival [38, 56, 57] information, all of which have been demonstrated in modern human populations. The lack of evidence for the omission bias might indicate the nonexistence of such a bias or the elimination of its effects in modern environments. Another possibility is that our between-subjects experiment design with a sample size of 202 participants might have been inadequate to detect a small or medium effect of omission bias. A replication of the current experiment with a larger sample size might help to clarify this possibility.

We also found no evidence to support the confirmation bias in either severity ratings or recall. This might be due to participants assessing the severity of symptoms independently of their causes. The lack of evidence for this hypothesis in recall is more difficult to explain. Research on the relationship between attitudes and recall has found contradictory results, both supporting and failing to support this hypothesis [58]. For instance, Frost et al. [42]found that people were better at recognising information that is congruent with their previous beliefs. In contrast, active resistance to arguments that are incongruent with previous beliefs has also been shown to improve recall [59].

One limitation of our test of the confirmation bias was related to the need to recruit participants with pre-existing pro- and anti-vaccination attitudes. Problematically, there was a mismatch between the attitudes towards vaccines self-reported in the pre-screening and the attitudes reported in the experiment, with fewer participants reporting anti-vaccination attitudes in the experiment than in the pre-screening. A possible explanation for this mismatch is that some participants assumed that reporting anti-vaccination attitudes would ensure their participation in a greater number of studies and, therefore, earn them more money [45]. Consequently, the use of pre-screening questions on participant recruiting websites (e.g. Prolific, Amazon Mechanical Turk) to collect responses from people with low frequency conditions such as anti-vaccination attitudes might not be the best option for further research. Future studies should create better procedures to recruit participants with anti-vaccination attitudes, for instance, by directly recruiting from anti-vaccination groups.

Another limitation is that participants had to imagine a hypothetical scenario in which their decision to vaccinate or not to vaccinate was randomly imposed by us. It is possible that the results might have been different if participants had to choose whether to vaccinate or not by themselves and later rate/recall the symptoms/side effects. Such a procedure may have increased the participants' investment in the task, providing more valid responses.

## Conclusion

The lack of convincing evidence for the omission bias suggests that the spread of anti-vaccination messages might not be explained by an omission bias acting against the practice of vaccination. Instead, the messages transmitted by people with anti-vaccination beliefs might spread due to having particular characteristics (e.g. being simple, concrete, emotional, unexpected, narrative, etc.) that make them especially "sticky" in human minds [60]. Importantly, the same characteristics of these messages might also be used to promote vaccination [20, 61], providing a more optimistic prospect for countering anti-vaccination information.

## Supporting information

**S1 File. Supplementary materials A-E.**
(DOCX)

**S2 File. Dataset for Pretest 1.**
(CSV)

**S3 File. Dataset for Pretest 2.**
(CSV)

**S4 File. Dataset for experiment.**
(CSV)

## Author Contributions

**Conceptualization:** Ángel V. Jiménez, Alex Mesoudi, Jamshid J. Tehrani.

**Formal analysis:** Ángel V. Jiménez.

**Funding acquisition:** Alex Mesoudi.

**Methodology:** Ángel V. Jiménez.

**Supervision:** Alex Mesoudi.

**Writing – original draft:** Ángel V. Jiménez, Alex Mesoudi, Jamshid J. Tehrani.

**Writing – review & editing:** Ángel V. Jiménez, Alex Mesoudi, Jamshid J. Tehrani.

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
