## [Decision Letter · Decision Letter 0]

24 Oct 2019

PONE-D-19-27807

NO EVIDENCE THAT OMISSION AND CONFIRMATION BIASES AFFECT THE PERCEPTION AND RECALL OF VACCINE-RELATED INFORMATION

PLOS ONE

Dear Mr Jimenez,

Thank you for submitting your manuscript to PLOS ONE. After careful consideration, we feel that it has merit but does not fully meet PLOS ONE’s publication criteria as it currently stands. Therefore, we invite you to submit a revised version of the manuscript that addresses the points raised during the review process.

We would appreciate receiving your revised manuscript by Dec 08 2019 11:59PM. To enhance the reproducibility of your results, we recommend that if applicable you deposit your laboratory protocols in protocols.io, where a protocol can be assigned its own identifier (DOI) such that it can be cited independently in the future. For instructions see: http://journals.plos.org/plosone/s/submission-guidelines#loc-laboratory-protocols

We look forward to receiving your revised manuscript.

Kind regards,

Catharine Penelope Cross

Academic Editor

PLOS ONE

2. Please provide additional details regarding participant consent. In the ethics statement in the Methods and online submission information, please ensure that you have specified what type of consent you obtained (for instance, written or verbal, and if verbal, how it was documented and witnessed). If your study included minors, state whether you obtained consent from parents or guardians. If the need for consent was waived by the ethics committee, please include this information.

Additional Editor Comments (if provided):

Dear Mr Jiménez

I have now received two very favourable reviews of your manuscript and have read it carefully myself. I think it is very close to being ready to publish.

Both reviewers made suggestions for improving the paper further, however, and I have one or two myself. I have complied these suggestions into an outline here and would also encourage you to read the original comments of both reviewers in full.

Additional citations:

R1 made some suggestions of papers on vaccine attitudes which you might wish to cite. I’d encourage you to have a look at these, though I’ll leave it up to you which ones you feel would really add to the paper. If R1 is right, however, the suggestion they make regarding a tweak to line 210 would be particularly relevant, as it would link to something directly relevant to your study.

Analysis:

R1 reported having some difficulty following your results section. I think it might be worth (if possible) adding a citation to similar analyses/guides to such analyses for readers not familiar with the AIC approach. In particular, I think the use of the word ‘logistic’ threw R1 off the trail a bit: it seems they believed you had tested a yes/no of whether each ppt recalled each item, rather than testing proportions recalled correctly (I went to confirm what you’d done by looking at the variable ‘Recall’ and its range in the supplementary material). It might be worth adding a clarifying sentence or two in the main text to walk readers through the analysis process a little more slowly.

Preregistration details:

R2 noted that pre-registration is a particular strength of the paper. They seem not to have located the scripts and data in supplementary material, but I think these will be much more findable when the paper is nicely formatted and online. I had a look and they seem clear and thorough to me.

R2 did note that the dates of data collection don’t seem to be given in the paper, and I can’t find them either. Please could you specify the period of data collection in the main text?

R2 also suggests that there should be a clearer marking of analyses that are and are not part of the pre-registered plan. I note that these are marked in the R scripts, but would ideally like to see a brief note of this in (the results section of) the main text as well.

Power

Finally, R2 noted the age-old problem of inferring a negative from a null result, and suggested that if the ‘true’ effect is small then N=202 might actually miss it. Please could you tweak the sentence that R2 notes in their review and any similar ones accordingly. An additional suggestion from me would be to discuss the range of plausible effect sizes according to the analysis, as well as reporting whether or not a predictor is part of a best-fitting model. Joe Stubbersfield does this in the main text and supplementary material of this paper: Stubbersfield, J. M., Dean, L. G., Sheikh, S., Laland, K. N., & Cross, C. P. (2019). Social transmission favours the ‘morally good’over the ‘merely arousing’. Palgrave Communications, 5(1), 3.

Gender of participants

I found the (quite interesting!) rationale for testing only female participants in the supplementary material, but I think it’s worth mentioning briefly in the main text because many readers will wonder about it.

I hope these suggestions are helpful. I’ll be happy to clarify any that require it. I look forward to receiving your revision – I think this paper will be a valuable addition to the literature.

Best wishes

Kate Cross

Reviewers' comments:

Reviewer's Responses to Questions

**Comments to the Author**

1. Is the manuscript technically sound, and do the data support the conclusions?

Reviewer #1: Yes

Reviewer #2: Yes

2. Has the statistical analysis been performed appropriately and rigorously? 

Reviewer #1: Yes

Reviewer #2: Yes

3. Have the authors made all data underlying the findings in their manuscript fully available?

Reviewer #1: Yes

Reviewer #2: No

4. Is the manuscript presented in an intelligible fashion and written in standard English?

Reviewer #1: Yes

Reviewer #2: Yes

5. Review Comments to the Author

Reviewer #1: I uploaded my review as an attachment.

I uploaded my review as an attachment.

I uploaded my review as an attachment. (Couldn't submit the manuscript if this section was left under 200 characters)

Reviewer #2: The manuscript is clear, the research well motivated and thoroughly carried out, the results presented clearly. It is important that these results are published.

I have three recommendations which I think are important to see the full value of this work realised

1. Regarding Pre-registration

The preregistration of the analysis is a great thing and the authors could make more of it (since it is a strength of an analysis with several DVs and multiple plausible analysis pathways for the exact analysis to be pre-declared).

Regardless of whether the authors decide to emphasise the pre-registration they should#

A Clearly distinguish pre-registered analyses from exploratory analyses in the results

B Clarify the timeline: the ethics for this project was given June 2016, the pre-registration is dated Dec 2017, but timestamped Feb 2018. When was the data collected? This information should be declared

2. Open Data

Why not make the data (and final analysis code) from this study available?

3. Risk of false negatives

The authors should discuss the possibility that their null results are false negatives. In particular, a statistical power analysis of a between subjects contrast will reveal that n=202 is not actually that many for detecting small to medium effects. The consequence of this must be to weaken the conclusion that this study is evidence of lack of omission bias (i.e. it elevates the plausibility that the study is merely a lack of evidence regarding the presence of an omission bias). So, for example, a sentence like this should be revised or qualified "The lack of evidence for the omission bias indicates the nonexistence of such a bias or the elimination of its effects in modern environments."

6. PLOS authors have the option to publish the peer review history of their article (what does this mean?). If published, this will include your full peer review and any attached files.

Reviewer #1: Yes: Sacha Altay

Reviewer #2: Yes: Tom Stafford

---

## [Author Response · Author response to Decision Letter 0]

10 Dec 2019

Thank you for your comments and suggestions. We have included a response to your comments on the Response to Reviewers. We also include our response here: 

# EDITOR

R1 made some suggestions of papers on vaccine attitudes which you might wish to cite. I’d encourage you to have a look at these, though I’ll leave it up to you which ones you feel would really add to the paper. If R1 is right, however, the suggestion they make regarding a tweak to line 210 would be particularly relevant, as it would link to something directly relevant to your study.

R1 is right: the study does not provide evidence to rule out the disgust-antivaccination link. Consequently, we have changed the sentence to be more specific to the omission bias: “the lack of convincing evidence for the omission bias suggest that the spread of anti-vaccination messages might not be explained by an omission bias acting against the practice of vaccination” (lines 535-537). We do not comment on the disgust-antivaccination link, as this is not the topic of our article. 

We have also added one of the suggested references on vaccine communication. We don’t think we need to specifically address the literature on vaccine communication as the article focuses on cognitive biases, not in specific interventions to make pro-vaccination messages more appealing. 

Analysis:

R1 reported having some difficulty following your results section. I think it might be worth (if possible) adding a citation to similar analyses/guides to such analyses for readers not familiar with the AIC approach. In particular, I think the use of the word ‘logistic’ threw R1 off the trail a bit: it seems they believed you had tested a yes/no of whether each ppt recalled each item, rather than testing proportions recalled correctly (I went to confirm what you’d done by looking at the variable ‘Recall’ and its range in the supplementary material). It might be worth adding a clarifying sentence or two in the main text to walk readers through the analysis process a little more slowly.

We have added a few sentences to explain how model comparisons work: “For our analyses of average severity ratings, we adopted a model comparison approach (Burhmam and Anderson, 2002). We first ran a null linear model that used a fixed intercept (i.e. the mean of the outcome variable) to predict each participants’ average severity ratings. We compared this null model with two linear regression models directly derived from our hypotheses using Akaike’s Information Criterion (AIC, see Table 2). A lower AIC by a difference of two or larger was considered evidence of better fit to the data” (Lines 382-391).

We have also added a reference to a classical book on the use of AIC for model comparisons and statistical inference: 

Burnham, K. P., & Anderson, D. R. (2002). Model Selection and Multi-Model Inference: A Practical Information-Theoretic Approach. Secaucus, US: Springer.

We have also added a few sentences to be clearer about our use of logistic regression for a proportional outcome and added a reference about how to conduct this type of regression model: “For our analyses of recall, we use the same model comparison approach as for the analyses of average severity ratings. As our outcome variable was the proportion of correctly recalled symptoms/side effects, we modelled the relationship between the predictors and the outcome variables using logistic regression models for a proportional outcome (Zuur, Ieno, Saveliev, Smitt & Walker, 2009, pp. 254-257). The fit of a null model was compared with our confirmatory models (see Table 3)” (Lines 414-419). 

R2 noted that pre-registration is a particular strength of the paper. They seem not to have located the scripts and data in supplementary material, but I think these will be much more findable when the paper is nicely formatted and online. I had a look and they seem clear and thorough to me.

The data and the R scripts were submitted together with the manuscript. We want the data and R scripts to be accessible for everybody together with the article on PlosOne’s website. 

R2 did note that the dates of data collection don’t seem to be given in the paper, and I can’t find them either. Please could you specify the period of data collection in the main text?

We have added a sentence (Lines 276-277) with the dates of data collection (5th March and 31st March 2018). 

R2 also suggests that there should be a clearer marking of analyses that are and are not part of the pre-registered plan. I note that these are marked in the R scripts, but would ideally like to see a brief note of this in (the results section of) the main text as well.

We have provided a more extended explanation of the changes conducted in the data analyses with respect to the preregistered analyses at the beginning of the Results section. Specifically, we report that we are not reporting regression models with interaction terms without main effect terms due to their controversial nature in statistics. We explain that instead we report regression models with both main effect and interaction terms. We also note that this change has not qualitatively affected the results and provide both analyses as supplementary materials so that the readers can check this by themselves.

Power

Finally, R2 noted the age-old problem of inferring a negative from a null result, and suggested that if the ‘true’ effect is small then N=202 might actually miss it. Please could you tweak the sentence that R2 notes in their review and any similar ones accordingly. An additional suggestion from me would be to discuss the range of plausible effect sizes according to the analysis, as well as reporting whether or not a predictor is part of a best-fitting model. Joe Stubbersfield does this in the main text and supplementary material of this paper: Stubbersfield, J. M., Dean, L. G., Sheikh, S., Laland, K. N., & Cross, C. P. (2019). Social transmission favours the ‘morally good’over the ‘merely arousing’. Palgrave Communications, 5(1), 3.

As suggested, we have qualified the statement “The lack of evidence for the omission bias indicates the nonexistence of such a bias or the elimination of its effects in modern environments” by changing it to “The lack of evidence for the omission bias MIGHT INDICATE the nonexistence of such a bias or the elimination of its effects in modern environments” and adding “Another possibility is that our between-subjects experiment design with a sample size of 202 participants might have been inadequate to detect a small or medium effect of omission bias. A replication of the current experiment with a larger sample size might help to clarify this possibility”(Lines 500-505). 

The best fitting model for average severity ratings is the null model, so the best-fitting model does not include any of the predictors of interest. For proportion of correct recall, the best fitting model did include our predictors of interest (Condition and Vaccination attitudes) and their interaction together with Reaction Time. However, the model fit was similar to the control model (with Reaction Time as unique predictor). Consequently, we do not think it is necessary to report this. 

Gender of participants

I found the (quite interesting!) rationale for testing only female participants in the supplementary material, but I think it’s worth mentioning briefly in the main text because many readers will wonder about it.

We have now included the sentences about our selection of only female participants in the main text (Lines 250-258)

---

## [Editor Report · Decision Letter 1]

27 Jan 2020

NO EVIDENCE THAT OMISSION AND CONFIRMATION BIASES AFFECT THE PERCEPTION AND RECALL OF VACCINE-RELATED INFORMATION

PONE-D-19-27807R1

Dear Dr. Jimenez,

We are pleased to inform you that your manuscript has been judged scientifically suitable for publication and will be formally accepted for publication once it complies with all outstanding technical requirements.

With kind regards,

Catharine Penelope Cross

Academic Editor

PLOS ONE

Additional Editor Comments (optional):

Dear Dr Jimenez

I am delighted to see such a clear and thorough response to the reviewer comments. Many thanks to you and your co-authors for making the changes and explaining the rationale for these changes (or absence of changes) in such an exemplary way.
---

## [Editor Report · Acceptance letter]

24 Feb 2020

PONE-D-19-27807R1 

No Evidence that Omission and Confirmation Biases Affect the Perception and Recall of Vaccine-related Information 

Dear Dr. Jiménez:

I am pleased to inform you that your manuscript has been deemed suitable for publication in PLOS ONE. Congratulations! Your manuscript is now with our production department. 

With kind regards,

on behalf of

Dr. Catharine Penelope Cross 

Academic Editor

PLOS ONE